# Fractal Properties of Heart Rate Dynamics: A New Biomarker for Anesthesia—Biphasic Changes in General Anesthesia and Decrease in Spinal Anesthesia

**DOI:** 10.3390/s22239258

**Published:** 2022-11-28

**Authors:** Jheng-Yan Lan, Jiann-Shing Shieh, Jia-Rong Yeh, Shou-Zen Fan

**Affiliations:** 1Department of Anesthesiology, Taipei Veterans General Hospital, Yuli Branch, Hualian 98142, Taiwan; 2Department of Mechanical Engineering, Yuan Ze University, Taoyuan 320, Taiwan; 3Department of Anesthesiology, National Taiwan University Hospital, Taipei 10002, Taiwan; 4Department of Anesthesiology, En Chu Kong Hospital, New Taipei City 237, Taiwan; 5College of Medicine, National Taiwan University, Taipei 10002, Taiwan

**Keywords:** depth of anesthesia (DOA), detrended fluctuation analysis (DFA), short-term scaling exponent (DFAα1), general anesthesia, spinal anesthesia

## Abstract

Processed electroencephalogram (EEG) has been considered a useful tool for measuring the depth of anesthesia (DOA). However, because of its inability to detect the activities of the brain stem and spinal cord responsible for most of the vital signs, a new biomarker for measuring the multidimensional activities of the central nervous system under anesthesia is required. Detrended fluctuation analysis (DFA) is a new technique for detecting the scaling properties of nonstationary heart rate (HR) behavior. This study investigated the changes in fractal properties of heart rate variability (HRV), a nonlinear analysis, under intravenous propofol, inhalational desflurane, and spinal anesthesia. We compared the DFA method with traditional spectral analysis to evaluate its potential as an alternative biomarker under different levels of anesthesia. Eighty patients receiving elective procedures were randomly allocated different anesthesia. HRV was measured with spectral analysis and DFA short-term (4–11 beats) scaling exponent (DFAα1). An increase in DFAα1 followed by a decrease at higher concentrations during propofol or desflurane anesthesia is observed. Spinal anesthesia decreased the DFAα1 and low-/high-frequency ratio (LF/HF ratio). DFAα1 of HRV is a sensitive and specific method for distinguishing changes from baseline to anesthesia state. The DFAα1 provides a potential real-time biomarker to measure HRV as one of the multiple dimensions of the DOA.

## 1. Introduction

Delivering adequate anesthetics to achieve a suitable depth of anesthesia (DOA) is an ongoing challenge for anesthesiologists. Researchers have focused on analyzing electroencephalograms (EEG) as reliable noninvasive ways to monitor the DOA [1]. However, measures derived from EEGs mainly reflect anesthetic effects on the cortex and consciousness state but not the brainstem and hemodynamic stability. The latter two are controlled by the autonomic nervous system (ANS) nuclei and are included in the multiple dimensions of DOA. Heart rate variability (HRV) has been used to assess the ANS [2,3,4,5,6] and may reflect the effects of the anesthetic on the brainstem component [7]. Propofol infusion [8,9,10,11], desflurane inhalation [12,13,14], and spinal anesthesia [15,16,17] have been reported to modulate the sympathovagal effects using spectral analysis of HRV. Because nonlinear phenomena are involved in the genesis of human heart rate fluctuations [18] and limitations of traditional linear measurement are observed, a new analytical method has been developed to evaluate cardiac regulation and to characterize the feature of heart rate (HR) dynamics [7,19,20,21,22].

Fractal property has been found in the biomarker of EEG and HR. Detrended fluctuation analysis (DFA), a nonlinear analytical technique, is a method designed not only to assess the magnitude of variability but also the quality, scaling, and fractal-like correlation properties of the signals [19,23,24,25,26]. The scaling exponents of HR are the slope of fluctuation (log) and window size (log). Values 1 to 1.5 represent noise with complexity, whereas values near 0.5 are less complex white noise. The short-term exponent α1 (4–11 beats) is strongly correlated with ANS and acute changing of cardiac function. The long-term exponent α2 (>12 beats) is correlated with chronic cardiac function [27]. A decreased exponent means increased randomness and represents the loss of cardiac responses to the external environment [4,23,28,29,30,31,32]. DFA α1, because of its quick response with fewer beats calculation, less dependence on HR, and robustness against artifacts, could be a sensitive biomarker of cardiovascular regulation intraoperatively.

Although changes in fractal characteristics of EEG as a measure of DOA have been published [33] and many studies using DFA and other nonlinear methods of HRV to estimate the cardiac function of patients and the physiological status of athletes are reported [27,34,35,36], little attention has been devoted to fractal characteristics of HRV during different anesthesia state. In the present study, we attempted to investigate the changes in DFA induced by intravenous propofol, inhalational desflurane, and spinal anesthesia and compared them with parameters derived from traditional spectral analysis.

## 2. Materials and Methods

### 2.1. Subjects and Study Protocol

After Institutional and Ethical investigational Committee approval (NTUH IRB: 200902012R) and obtaining informed consent, we enrolled 80 healthy (ASA 1) adult patients scheduled for orthopedic or general surgery under either spinal or general anesthesia in our study. Patients with a history of ischemic heart disease, congestive heart failure, diabetes mellitus, or any other disorders known to affect autonomic nervous functions were excluded. None of the patients were taking medications that may affect cardiovascular functions.

### 2.2. General Patient Management

Each patient was fasted at least 8 h prior to testing. Vigorous exercise, alcohol, or coffee intake were prohibited for 48 h before the scheduled surgery. Without any premedication, on the day of surgery, the patient lay in a supine position in a quiet room at least 5 min prior to preanesthetic baseline data collection [37]. A baseline electrocardiogram (ECG) was recorded and stored on a personal computer by ECG Holter (E30-8010, Micro-Star International Co., New Taipei City, Taiwan). The sampling rate was 1000 Hz. An AS/3 Anesthesia Monitor (Datex-Engstrom, Finland) was used to collect blood pressure (BP) data, ECG, and pulse oximetry (SpO_2_) signals from the patients. Randomization was achieved using an opaque sealed envelope technique that had been sorted by a computer-generated random allocator. Patients undergoing elective trunk or upper limb surgery were randomly allocated to two groups receiving total intravenous infusion with propofol (Group P, *n* = 20) or inhalational desflurane anesthesia (Group D, *n* = 20). DOA was continuously monitored using the A-line Index (AAI) generated by an auditory evoked potential (AEP) monitor (Danmeter, Odense, Denmark). AAI value below 35 is considered deep-plane anesthesia, and between 35 and 60, light-plane anesthesia. AEP was elicited with a binaural click stimulus of 65 dB intensity, 2 ms duration, and a repetition rate of 9 Hz. Patients undergoing elective lower limb surgery were randomly allocated to two groups receiving intrathecally either low-dose hyperbaric bupivacaine alone (Group LM, *n* = 20) or bupivacaine supplemented with intrathecal fentanyl (Group LMf, *n* = 20).

### 2.3. Induction and Maintenance of Intravenous Propofol Anesthesia

All patients received 100% oxygen via a facemask for 2 to 3 min prior to induction of anesthesia. Anesthesia was induced and maintained with continuous infusion of propofol at a rate of 300 μg·kg^−1^ min^−1^ via an indwelling peripheral venous catheter. Spontaneous respiration was maintained and assisted with gentle intermittent positive-pressure ventilation (IPPV) via a mask if required to maintain End-tidal carbon dioxide (ETCO_2_) between 30 and 40 mmHg.

### 2.4. Induction and Maintenance of Desflurane Anesthesia

To avoid possible bronchial irritation caused by desflurane, all patients received intravenous fentanyl 1μg/kg before induction. Anesthesia was induced with an incremental increase in desflurane of 3, 6, 9, and 12% in a gas mixture of 50:50% O_2_ and N_2_O. Spontaneous ventilation was maintained and assisted with IPPV via mask if required to maintain ETCO_2_ between 35 and 40 mmHg.

### 2.5. Spinal Anesthesia

Spinal anesthesia was performed with patients placed in a lateral decubitus position with a 27-G needle inserted via L3-4 or L4-5 interspace. In Group LM, 1.2 mL of 0.5% hyperbaric bupivacaine was injected intrathecally; in Group LMf, 1.2 mL of 0.5% hyperbaric bupivacaine combined with 20 μg (0.4 mL) of intrathecal fentanyl was injected (total injectant was 1.6 mL). The block height was determined by pinprick and cold sensation test examined every 10 min until desired level T6-7 was reached. Patients who developed severe bradycardia (HR < 50/min) or hypotension (systolic BP < 80 mmHg) after spinal anesthesia requiring treatment with atropine or ephedrine were excluded from the final statistical analysis.

### 2.6. Data Collection

The recorded ECG signals were retrieved after surgery to measure the consecutive R–R intervals using LabChart v5 (ADInstruments, Colorado Springs, CO, USA), and all R–R intervals were edited manually to exclude all premature beats and noise. The last 660 stationary R–R intervals were obtained for DFA and spectral analysis. If the percentage of deletion was greater than 5%, the subject was excluded from the study.

### 2.7. Time and Frequency Domain Analysis

The parameters of HRV were calculated according to the Task Force of the European Society of Cardiology and the North American Society of Pacing and Electrophysiology [37]. The time domain measures of HRV, which included mean R–R intervals, the standard deviation of normal-to-normal (N–N) interbeat intervals (SDNN), and the root mean square of the successive difference in N–N intervals (RMSSD), were calculated. The direct current component was excluded before the calculation of power spectra. The areas under the spectra peaks within the ranges of 0.04–0.15 Hz and 0.15–0.4 Hz were defined as low-frequency power (LFP) and high-frequency power (HFP), respectively. Normalized high-frequency power (nHFP = 100 × HFP/(LFP + HFP)) was used as the index of cardiac vagal modulation, and the normalized low-frequency power (nLFP = 100 × LFP/(LFP + HFP)) as the index of combined sympathetic and vagal modulation (Figure 1).

### 2.8. Detrended Fluctuation Analysis (DFA)

The total time series was first integrated and divided into segments of length n; each segment was then detrended by subtracting the best linear fit. The fluctuation function F(n) was then calculated as the root mean square of the detrended time series as a function of the segment size n. The α value represents the correlation properties of the time series (Figure 2) [30,38].

### 2.9. Statistical Analysis

Statistical analysis was performed with STATA 8.2 (StataCorp, College Station, TX, USA). We decided that a 10% difference in percentage changes of HRV parameters relative to baseline between the groups is important; therefore, *n* = 15 patients in each group would be necessary to detect such a difference if α = 0.05 and β = 0.1. The normality of distribution was tested using Shapiro–Wilk test. Because of the data’s skewed distribution, SDNN, low-frequency/high-frequency power ratio (LF/HF ratio), nLFP, and nHFP were transformed by calculating their neutral logarithm. The differences within groups were analyzed by multivariate analysis of variance (MANOVA) with Bonferroni correction of multiple measurements. All data are expressed as the mean ± SD.

A Pearson product-moment correlation analysis was used if the data analyzed were normally distributed, or a Spearman’s rank order correlation analysis was used if the data analyzed were distribution-free data in the correlation analysis between traditional measures and DFA measures.

A ROC curve was used to estimate the sensitivity and specificity of the different analysis methods in individual anesthesia states. The ROC curve reflects relative true positive, true negative, false positive, and false negative values termed specificity and sensitivity. The area under the ROC curve measures discrimination, that is, the ability of the analytical methods to correctly classify HR series data before and after anesthesia. Thus, the purpose of this comparison is to examine whether DFAα1 and LF/HF ratio would exhibit different sensitivities and specificities under different types of anesthesia.

## 3. Results

### 3.1. Baseline Characteristics of the Study Subjects

In the general anesthesia groups, two patients in Group D were excluded from the final statistical analysis because of excitement and arrhythmia following the induction of anesthesia. Demographic data were listed in Table 1. Under deep anesthesia with propofol (AAI < 35), BP decreased significantly, while HR remained stable (Table 1), whereas, with desflurane, both HR and BP decreased significantly from baseline (Table 1). In the spinal anesthesia groups, no patient received sedatives nor required atropine treatment. No patient had arrhythmia >5% throughout the study period. All 40 patients enrolled in the study were included in the final statistical analysis. Thirty minutes after induction of spinal anesthesia, HR decreased significantly in both groups, and BP decreased significantly only in group LMf, but not in group LM (Table 1).

### 3.2. HR Variability and DFAα1 between Baseline and after Anesthesia

Spectral and DFAα1 analysis of HRV parameters at baseline and postanesthesia are shown in Table 2.

Because of the data’s skewed distribution, SDNN, LF/HF ratio, nLFP, and nHFP were transformed by calculating their neutral logarithm. The differences within groups were analyzed by multivariate analysis of variance with Bonferroni correction of multiple measurements.

Group LM, spinal anesthesia with only bupivacaine; Group LMf, spinal anesthesia supplemented with intrathecal fentanyl; Group P, intravenous anesthesia with propofol infusion; Group D, inhalational anesthesia with desflurane.

SA—spinal anesthesia; AAI—A-Line ARX Index; SDNN—standard deviation of normal-to-normal interbeat intervals (N–N intervals); RMSSD–root mean square of successive differences in N–N intervals; TP—total power; LFP—low-frequency power; HFP—high-frequency power; nLFP—normalized low-frequency power; nHFP—normalized high-frequency power; LF/HF ratio—low-frequency/high-frequency ratio; DFAα_1_—detrended fluctuation analysis short-term scaling exponent.

General anesthesia: in both propofol and desflurane groups, DFAα1 increased significantly under light-plane anesthesia (35 < AAI < 60) but decreased significantly under the deep plane (AAI < 35) (Figure 3 and Figure 4). LF/HF ratio, nLFP, and LFP decreased, whereas SDNN and nHFP increased significantly from baseline in Group D under the deep-plane anesthesia (AAI < 35). RMSSD, HFP, nHFP, and LFP decreased significantly in Group P under the deep-plane anesthesia when AAI < 35 (Table 2).

Spinal anesthesia: following spinal anesthesia, nLFP, LF/HF ratio, and DFAα1 decreased significantly, whereas nHFP increased significantly from baseline in both groups (Figure 5 and Figure 6). SDNN decreased significantly only in Group LMf, and HFP decreased significantly only in Group LM (Table 2).

### 3.3. Correlation between α1 and Other HR Variability Variables

DFAα1 value did not correlate with the traditional LF/HF ratio in all groups following induction of anesthesia (Table 3). RMSSD in the time domain, nLFP, nHFP, and HFP in the frequency domain correlated significantly with DFAα1 in Group P under deep-plane anesthesia (AAI < 35). DFAα1 also significantly correlated with nLFP and nHFP in Group LMf after spinal anesthesia.

### 3.4. The ROC Curve

In the area under the curve (AUC) in spinal anesthesia Group LM and LMf, the AUC of DFAα1 were 0.84 and 0.98, respectively; the AUC of LF/HF ratio was 0.65 and 0.73, respectively (Figure 7A). In general anesthesia Groups P and D, the AUC of DFAα1 was 0.68 and 0.85, respectively; the AUC of LF/HF ratio was 0.55 and 0.72, respectively (Figure 7B,C).

## 4. Discussion

In this study, we demonstrated the biphasic effects of HRV after general anesthesia induction with desflurane or propofol using DFA analysis compatible with the clinical “paradoxical excitation” phenomenon [39,40]. Biphasic effects, that is, the increase in DFAα1 followed by a decrease in DFAα1 at higher concentrations during the transition from awake to deep anesthesia, were not observed by traditional spectral HRV methods.

A biphasic reaction during induction of propofol anesthesia was also observed in spectral EEG analyses [41,42], i.e., a paradox excitation of EEG alpha and beta power even if unconsciousness was observed. Interestingly, nonlinear EEG methods do not follow a biphasic course but adequately reflect the hypnotic state [43]. Intravenous propofol infusion is known to cause a reduction in BP and HR and inhibit sympathetic [8,9] or parasympathetic [10,11] activity. This inhibition of autonomic activity is believed to be one of the major mechanisms underlying propofol-induced hemodynamic depression. Although it is generally agreed that propofol anesthesia is associated with a reduction in HRV [11,44,45], results regarding the effect on cardiac sympathetic or parasympathetic tone are conflicting. In the present study, we found that DFAα1 of HRV increased from the awake state to light general anesthesia (strong fractal property) and decreased from light to deep anesthesia (breakdown of fractal property). Tulppo et al. demonstrated changes in HR dynamics toward stronger short-term fractal correlation properties (α1) when ANS was activated, i.e., by decreasing vagal tone while increasing sympathetic activity, by physical stress [46]. Therefore, under light general anesthesia, increased DFAα1 is most likely due to predominant inhibition of vagal tone (decreased HFP) and concomitant inhibition but of slight, sympathetic tone. Such similar behavior of α1 has also been reported by many other physical stress studies, such as passive head-up tilt test [47], light-intensity exercise [48], and cold hand test [46]. As anesthesia becomes deeper, further inhibition of vagal and sympathetic tone occurs, causing DFAα1 to decrease. Traditional spectral analysis using LF/HF ratio did not reflect these significant changes in HRV induced by the deepening of anesthesia.

Biphasic changes, increase then decrease, in DFAα1 was also observed in desflurane anesthesia. Desflurane can cause neurocirculatory excitation manifested as an increase in HR and BP (increase in sympathetic activity) during induction and transitional stage as desflurane concentration increases [12,13,14]. In our study, under light-plane anesthesia, we did not see evidence of activation of sympathetic tone either by direct physical signs or by traditional spectral analysis of HRV; DFAα1, however, seemed able to detect the subtle changes in R–R interval dynamics better than traditional spectral analysis (Figure 3). Under deep anesthesia, both DFAα1 and LF/HF ratio decrease, which would indicate further inhibition of sympathetic tone (reduction in LFP).

The study showed DFAα1 decrease by the breakdown of the short-term fractal correlation properties of HR dynamics towards a more random pattern in spinal anesthesia. We found DFAα1 decrease is accompanied by the reciprocal increase in nHFP and a decrease in nLFP and LF/HF ratio. It also indicates that reciprocal change is associated with the breakdown of HR fractal property. The results showed enhanced vagal activity and withdrawal of sympathetic activity after spinal anesthesia had been observed in HRV [49]. Decreased DFAα1 has been observed in various disease conditions or under certain physical states, such as heart failure [23], before the onset of atrial fibrillation [29] and under cold face tests in healthy subjects [46]. However, when the local anesthetic was supplemented with fentanyl, we did not observe the synergetic effect of the sympathetic blockade as previously reported when fentanyl was intravenously administered [50]. We added 20 μg fentanyl (0.4 mL) to 1.2 mL of bupivacaine, increasing the injecting volume to 1.6 mL in patients in Group LMf. This could have made the local anesthetic less hyperbaric or even become hypobaric. However, Patterson et al. [51] have reported that the addition of fentanyl does not alter the extent of the spread of intrathecal isobaric bupivacaine. Furthermore, the blockade levels reached were equal between the two groups; we believe the increased volume and change in baricity of the injectate would have a very minimal impact on the result of our findings. The effect of intrathecal fentanyl might be confounded by bupivacaine or by its central nervous system sedative effects [52,53]. The difference in heart fractal property demonstrated by intrathecal and intravenous administration of fentanyl needs to be further investigated.

The present study did not show parameters derived from traditional spectral analysis of HRV correlate well with the DFAα1 value. The traditional spectral analysis measures the magnitude of each frequency component (oscillation mode), but the DFA method focuses on how this oscillation mode is arranged. Thus theoretically, DFA scaling exponents will not correlate with most HRV parameters derived from spectral analysis. The weak correlation during anesthesia may partially be due to uncontrolled breathing rate since some studies have reported a high correlation between DFA and LF/HF ratio in controlled ventilation [47,48]. However, this is still unclear and is a subject of controversy, as some studies are reporting a low correlation even with controlled ventilation [54] or a high correlation without controlled ventilation [55,56]. In our study, the rates of spontaneous breathing were similar between spinal and general anesthesia groups, 10–15 breaths/min under spinal and 8 −15 breaths/min under general anesthesia. This seemed to suggest other factors, such as differences in frequency characteristics, stationary assumption of spectral analysis, and BP may be involved in the inconsistency. In addition to respiratory rate, other respiratory parameters such as tidal volume [57] and possible mental stress during controlled breathing could also be important factors that affect the HRV.

Several nonlinear methods for HRV have tried to distinguish awake from sleep state under anesthesia [58] and detect subtle sympathetic and parasympathetic-mediated alteration [7]. In our study, we suggest that DFAα1 in HRV is a sensitive and specific index to be used in clinical practice. Though this method could not provide direct information regarding the modulation of ANS, it may reflect the response of ANS, which may be considered one of the multiple dimensions of DOA. Moreover, our method provides real-time measurement of DOA (less than 1 s), which is an indispensable requisite for monitoring.

There are several limitations of our study. First, we did not include patients with heart disease or heart failure because we wanted to investigate how anesthetic drugs interact with HRV without avoidable confounding factors [23,25,28,29]. Second, anesthesia-induced changes in respiratory rate and tidal volume could have influenced HRV. Thus, we would have underestimated the effects of anesthetics on HRV should induction of anesthesia have resulted in a decrease in both respiratory rate and tidal volume. Third, spinal anesthesia is accompanied by significant sedation [49,59,60]. Thus, both direct blockade of cardiac sympathetic fibers and sedation caused by spinal anesthesia might have exaggerated the sympathovagal effect of spinal anesthesia. Fourth, despite equivalent AAI value numbers, it is uncertain whether the depths of anesthesia achieved were equal between propofol and desflurane anesthesia. However, there is no convincing evidence that the AAI index number was independent of the DOA in our study. Since the AAI value falls in a similar pattern as anesthesia deepened with these two agents, we can reasonably believe that the DOA achieved was equal.

In the future, we plan to employ the DFA method to study HRV behavior in heart disease patients and other anesthetics, such as ketamine or opioids, as both agents are known to have very different effects on the central nervous system and/or autonomic nervous system modulation of the cardiovascular system.

## 5. Conclusions

Biphasic effects of HRV followed by deepening propofol or desflurane anesthesia were observed. DFAα1 decreased after spinal anesthesia, even with intrathecal bupivacaine alone or supplemented with fentanyl. Light general anesthesia resulted in the change in HR dynamics toward stronger short-term fractal correlation properties, while deep general anesthesia resulted in the breakdown of the fractal property. DFAα1 change may reflect the indirect response of ANS and provide a potential real-time nonlinear method to measure HR dynamic and DOA.

## Figures and Tables

**Figure 1 sensors-22-09258-f001:**
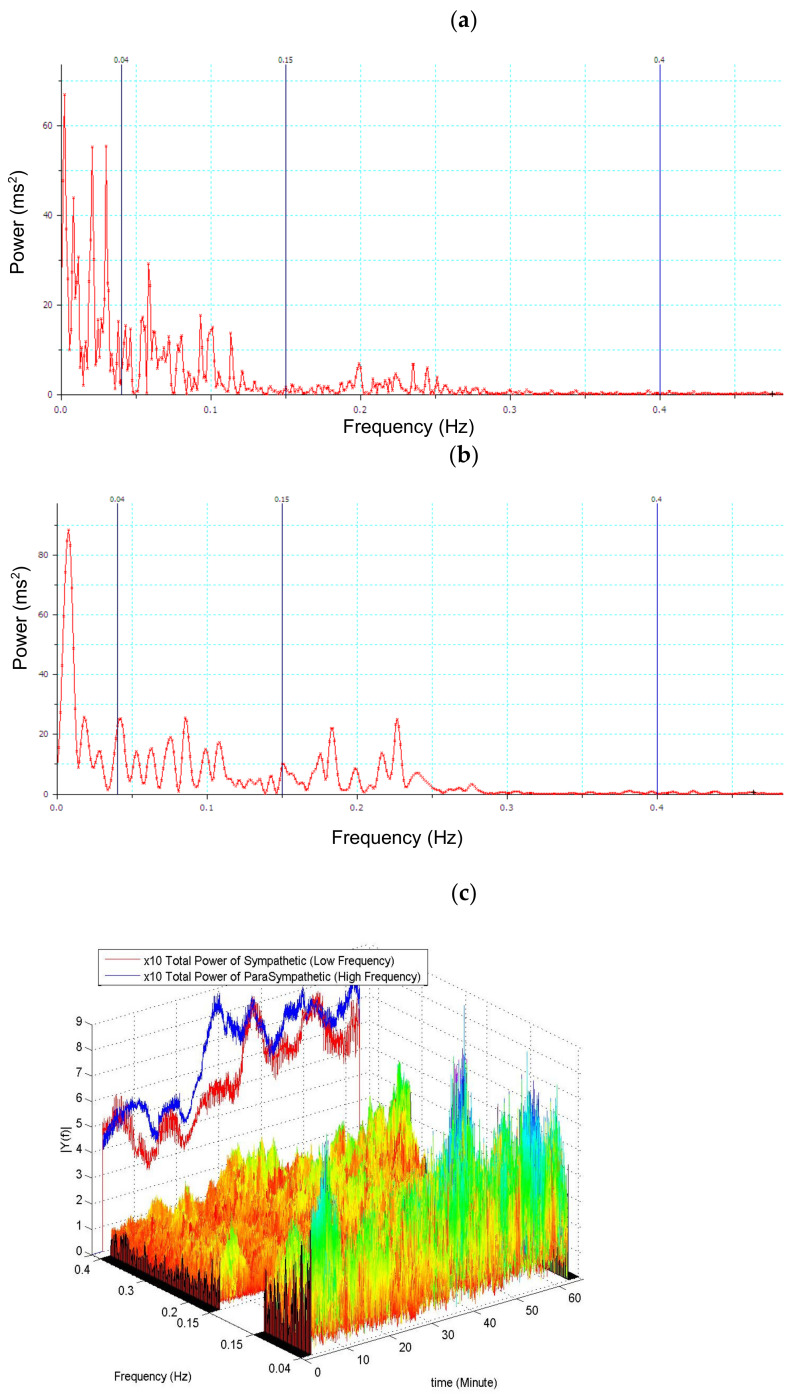
An example of spinal anesthesia R-R interval spectral analysis. (**a**) Without any premedication, on the day of surgery, the patient lay in a supine position in a quiet room at least 5 min prior to preanesthetic baseline data collection. (**b**) Shifting of power by R-R interval spectral analysis 30 min after spinal anesthesia. (**c**) The sliding window of 60 s shows the shifting power of LF and HF after spinal anesthesia.

**Figure 2 sensors-22-09258-f002:**
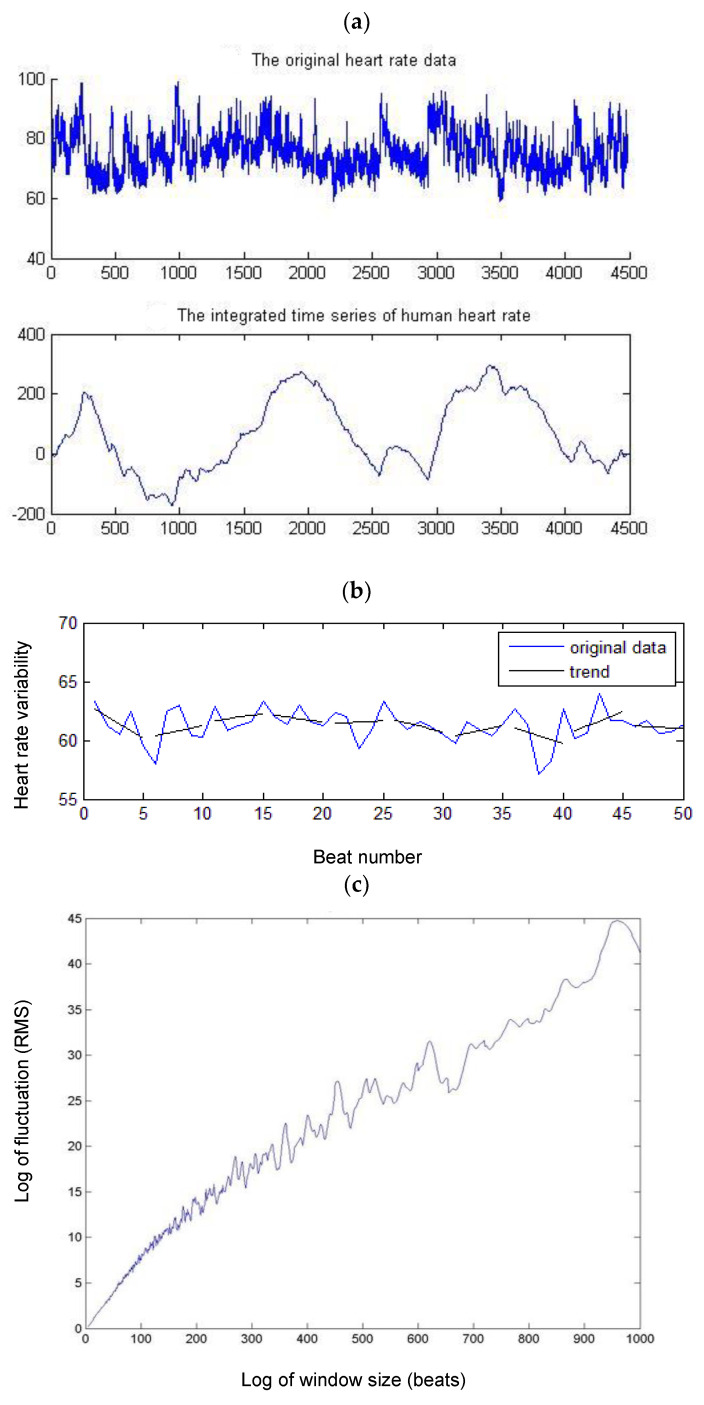
Calculation of DFA. (**a**) Original R–R interval and integrated time series. (**b**) The total time series was divided into segments; each segment was then detrended by subtracting the best linear fit F(n). (**c**) Root mean square of the detrended time series as a function of the segment size n. (**d**) If the time series is self-similar, a relationship indicates the presence of power law (fractal) scaling. The scaling exponent α can be estimated by a linear fit on the log-to-log plot of F(n) versus n. The α value represents the correlation properties of the time series. The global scaling exponent a value was calculated within the range of n, between *n* = 4 and *n* = 165 space. Short-term α1 was calculated between *n* = 4 and *n* = 11.

**Figure 3 sensors-22-09258-f003:**
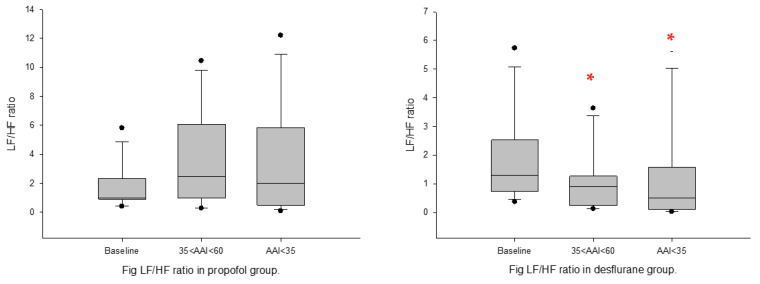
LF/HF ratio at baseline and post-general anesthesia for Group P (propofol); Group D (desflurane). Box plots show the median, 10th, 25th, 75th, and 90th percentiles as vertical boxes with error bars and plot all data points that lie outside the 10th and 90th percentiles as black circle dots. The significance levels with paired *t*-tests between baseline and successive measurement periods are as follows: * *p* < 0.05.

**Figure 4 sensors-22-09258-f004:**
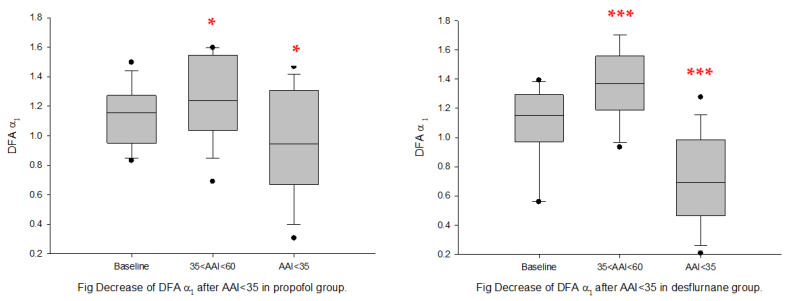
DFA_α1_ ratio at baseline and post-general anesthesia for Group P (propofol); D: Group D (desflurane). Box plots showing the median, 10th, 25th, 75th, and 90th percentiles as vertical boxes with error bars and plot all data points outside the 10th and 90th percentiles as black circle dots. The significance levels with paired *t*-tests between baseline and successive measurement periods are as follows: * *p* < 0.05; *** *p* < 0.001.

**Figure 5 sensors-22-09258-f005:**
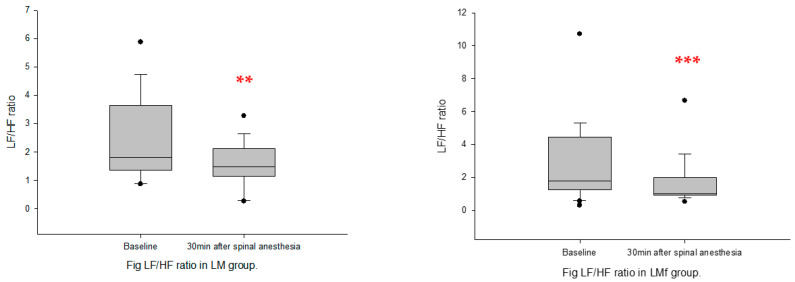
LF/HF ratio at baseline and post-spinal anesthesia. (**A**): Group LM; (**B**): Group LMf; Box plots showing the median, 10th, 25th, 75th, and 90th percentiles as vertical boxes with error bars and plot all data points outside the 10th and 90th percentiles as black circle dots. The significance levels with paired *t*-tests between baseline and successive measurement periods are as follows: ** *p* < 0.01; *** *p* < 0.001.

**Figure 6 sensors-22-09258-f006:**
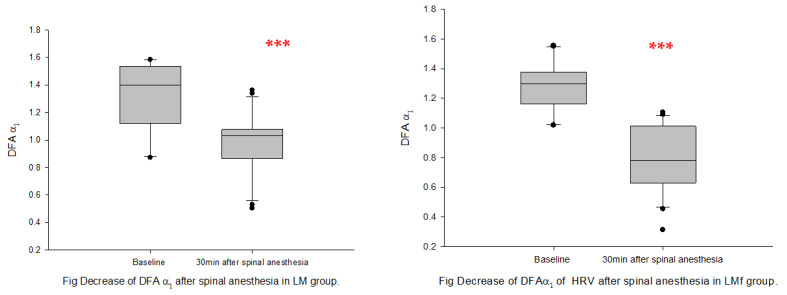
DFAα1 at baseline and post-spinal anesthesia. (**A**): Group LM; (**B**): Group LMf; Box plots showing the median, 10th, 25th, 75th, and 90th percentiles as vertical boxes with error bars and plot all data points outside the 10th and 90th percentiles as black circle dots. The significance levels with paired *t*-tests between baseline and successive measurement periods are as follows: *** *p* < 0.001.

**Figure 7 sensors-22-09258-f007:**
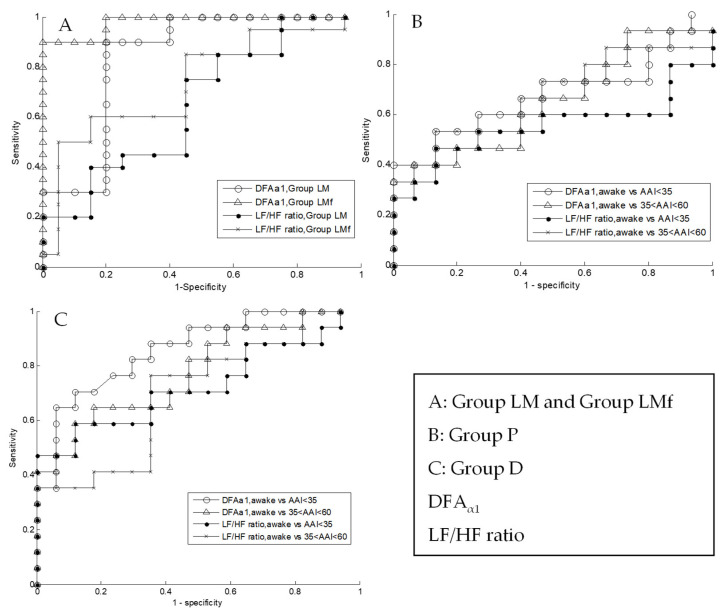
Receiver-operating curves analyses of DFA_α1_ and LF/HF ratio at baseline and postspinal (**A**) and general anesthesia (**B**,**C**) showing fractal analysis of HRV provided a more sensitive and specific information than traditional spectral measurement.

**Table 1 sensors-22-09258-t001:** Demography, hemodynamic baseline and postanesthesia data.

	Group LM (N = 20)	Group LMf (N = 20)	Group P (N = 20)	Group D (N = 18)
Age, yrs	41 ± 7	42 ± 8	42 ± 11	38 ± 12
Height, cm	164 ± 8	165 ± 8	159 ± 9	156 ± 8
Weight, kg	67 ± 10	66 ± 15	58 ± 11	53 ± 10
Sex, male/female	9/11	10/10	11/9	9/9
Block height: 30 min ater SA	T7 ± 2	T6 ± 3	-	-
BP (systolic)mmHg Baseline	130 ± 18	133 ± 15	123 ± 14	123 ± 16
30 min after SA or AAI < 35	120 ± 9 *	119 ± 15 *	106 ± 13 *	116 ± 9 *
BP (diastolic)mmHg Baseline	81 ± 9	80 ± 7	75 ± 9	77 ± 10
30 min after SA or AAI < 35	76 ± 9 *	72 ± 7 *	63 ± 8 *	69 ± 9
Heart rate Baseline	74 ± 13	72 ± 10	73 ± 11	76 ± 10
30 min after SA or AAI < 35	70 ± 12	65 ± 11 *	75 ± 9	56 ± 9 *

Unless otherwise noted, values represent the number or mean ± SD; * *p* < 0.05 vs. baseline; *p* values were calculated based on paired *t*-test; Group LM is spinal anesthesia with only bupivacaine; Group LMf is spinal anesthesia supplemented with intrathecal fentanyl; Group P is intravenous anesthesia with propofol infusion; Group D is inhalational anesthesia with desflurane; SA—spinal anesthesia; AAI—A-Line ARX Index; T—thoracic dermatome.

**Table 2 sensors-22-09258-t002:** Baseline and postanesthesia HRV parameters assessed by spectral analysis and DFA for all groups; (a) spinal anesthesia and (b) general anesthesia.

(a) Spinal Anesthesia						
Variable	Group LM			Group LMf		
	Baseline	30 min after SA		Baseline		30 min after SA
SDNN, ms	41.47 ± 22.25	50.26 ± 36.58		58.43 ± 33.34		44.24 ± 20.77 *
RMSSD, ms	25.89 ± 18.61	40.73 ± 45.05		34.23 ± 26.49		29.41 ± 14.01
TP, ms^2^/Hz	1893.32 ± 2028.8	3597.4 ± 5888.4		3359.34 ± 3237.51		2116.05 ± 1916.81
LFP, ms^2^/Hz	406.39 ± 407.40	843.68 ± 1119.11		727.41 ± 726.12		538.82 ± 647.51
HFP, ms^2^/Hz	290.73 ± 480.77	1418.3 ± 3331.6 *		444.99 ± 567.92		396.28 ± 435.91
nLFP	66.09 ± 11.94	55.92 ± 15.76 *		65.19 ± 17.80		56.03 ± 13.52 **
nHFP	33.91 ± 11.94	44.03 ± 15.76 *		34.81 ± 17.80		43.97 ± 13.52 **
LF/HF ratio	2.41 ± 1.46	1.52 ± 0.80 **		1.03 ± 0.07		0.95 ± 0.20 ***
DFAα_1_	1.32 ± 0.25	0.98 ± 0.21 ***		1.28 ± 0.17		0.80 ± 0.21 ***
(b) General anesthesia						
Variable	Group P			Group D		
	Awake	35 < AAI < 60	AAI < 60	Awake	35 < AAI < 60	AAI < 60
SDNN, ms	50.57 ± 22.20	53.05 ± 20.32	43.17 ± 20.29	45.22 ± 22.14	84.19 ± 27.02 **	63.94 ± 38.07 *
RMSSD, ms	30.52 ± 18.04	25.66 ± 14.90	24.26 ± 15.62 *	28.23 ± 17.37	40.79 ± 24.12	30.27 ± 20.81
TP, ms^2^/Hz	2511.51 ± 2286.57	2792.54 ± 2316.49	1676.45 ± 1647.79	2069.65 ± 2124.37	6318.92 ± 6019.69 **	3714.95 ± 4760.33
LFP, ms^2^/Hz	486.17 ± 373.34	652.74 ± 664.81	268.63 ± 293.54 *	480.40 ± 432.50	794.66 ± 1450.99	125.89 ± 139.98 **
HFP, ms^2^/Hz	476.05 ± 575.26	277.58 ± 344.17 **	282.01 ± 439.50	434.20 ± 606.60	1022.46 ± 1185.62	500.33 ± 613.01
nLFP	55.86 ± 16.75	62.51 ± 24.26 *	58.49 ± 28.97	57.34 ± 17.86	38.11 ± 19.64 **	36.12 ± 27.66 **
nHFP	44.14 ± 16.75	29.60 ± 16.72 *	41.51 ± 28.96	42.66 ± 17.86	51.37 ± 20.35	63.81 ± 27.66 **
LF/HF ratio	1.74 ± 1.50	3.68 ± 3.27	3.46 ± 3.76	1.90 ± 1.57	1.07 ± 1.03 *	1.15 ± 1.66 *
DFAα_1_	1.14 ± 0.21	1.26 ± 0.28 *	0.94 ± 0.35 *	1.10 ± 0.26	1.36 ± 0.25 ***	0.7 ± 0.31 ***

Values are mean ± SD; * *p* < 0.05; ** *p* < 0.01; *** *p* < 0. 001 vs. baseline.

**Table 3 sensors-22-09258-t003:** Correlation coefficient between traditional measures and DFAα_1_ after spinal (a) and general anesthesia for all groups (b).

(a) Spinal Anesthesia						
Variable	Group LM			Group LMf		
	Baseline		30 min after SA	Baseline		30 min after SA
DFAα1	-		-	-		-
SDNN	−0.3552		−0.1239	0.0589		0.0052
RMSSD	−0.0910		−0.1006	0.1257		0.1863
TP	−0.3399		0.1774	0.0749		0.1482
LFP	−0.1390		−0.2553	0.0867		0.2559
HFP	0.0325		0.5494 *	0.1975		−0.4815
nLFP	−0.0522		−0.5134 *	−0.1992		0.4759
nHFP	0.2895		0.4640	0.3009		−0.2979
LF/Hf ratio	−0.1906		−0.1101	0.1249		0.2952
(b) General anesthesia						
Variable	Group P			Group D		
	Awake	35 < AAI < 60	AAI < 60	Awake	35 < AAI < 60	AAI < 60
DFAα_1_	-	-	-	-	-	-
SDNN	−0.5132 *	−0.6323 *	−0.6005 *	−0.1023	0.1859	−0.5837
RMSSD	−0.3072	−0.2343	−0.1986	−0.0299	0.4139	−0.4375
TP	−0.1589	−0.0069	−0.0510	0.0073	0.1309	0.1047
LFP	−0.4991	−0.5600 *	−0.6626 *	−0.1595	0.3007	−0.6005
HFP	0.6678 *	0.4197	0.6630 *	0.2153	−0.1232	0.4172
nLFP	−0.6610 *	−0.4100	−0.6616 *	−0.2180	−0.0865	−0.4041
nHFP	0.5499 *	0.2995	0.3760	0.2881	0.0036	0.4661
LF/Hf ratio	−0.4214	−0.4171	−0.0876	−0.0084	0.2088	−0.4369

* Correlation coefficient: Pearson product-moment correlation analysis for normally distributed data and Spearman rank order correlation analysis for distribution-free data; Abbreviations: see Table 2.

## Data Availability

Not applicable.

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
