# Peer review of "Fractal Properties of Heart Rate Dynamics: A New Biomarker for Anesthesia—Biphasic Changes in General Anesthesia and Decrease in Spinal Anesthesia"

_sensors, 2022, doi:10.3390/s22239258_

Round 1

Reviewer 1 Report

The interpretation of the clinical significance of DFA is quite controversial in the literature. In this study, the authors propose the use of the alpha 1 index as a sensitive marker for anesthetic conditions, but do not characterize the reference values for alpha 1 and alpha 2. They also do not explain the pathophysiological interpretation of the observed changes. In addition, the figures are not very explanatory, making it difficult for the reader to understand. There is a need to improve the quality of information.

Author Response

We thank reviewer for the detailed review and comments, which is responded point by point below:

The interpretation of the clinical significance of DFA is quite controversial in the literature.

Although DFA is still not so clear in patients during anesthesia, many studies using DFA and other non-linear methods of HRV to estimate cardiac function of patients and the physiological status of athletes in recent years. [1-5]

In this study, the authors propose the use of the alpha 1 index as a sensitive marker for anesthetic conditions, but do not characterize the reference values for alpha 1 and alpha 2.

They also do not explain the pathophysiological interpretation of the observed changes.

We rephrase the introduction and add more references about the application DFA α1 to explain their variation of α1 and α2 index and explain their pathophysiological implication. We also try to explain why we use the α1 index as a possible sensitive biomarker intraoperatively.

In addition, the figures are not very explanatory, making it difficult for the reader to understand. There is a need to improve the quality of information.

We replace the line graphs with clearer box plots for more explanatory and also highlight DFA α1 is more sensitive than traditional spectral analysis in the manuscript.

  1. Mizobuchi, A.; Osawa, K.; Tanaka, M.; Yumoto, A.; Saito, H.; Fuke, S. Detrended fluctuation analysis can detect the impairment of heart rate regulation in patients with heart failure with preserved ejection fraction. J Cardiol 2021, 77, 72-78, doi:10.1016/j.jjcc.2020.07.027.
  2. Cui, X.; Tian, L.; Li, Z.; Ren, Z.; Zha, K.; Wei, X.; Peng, C.K. On the Variability of Heart Rate Variability-Evidence from Prospective Study of Healthy Young College Students. Entropy (Basel) 2020, 22, doi:10.3390/e22111302.
  3. Kawakami, D.M.O.; Bonjorno-Junior, J.C.; da Silva Destro, T.R.; Biazon, T.; Garcia, N.M.; Bonjorno, F.; Borghi-Silva, A.; Mendes, R.G. Patterns of vascular response immediately after passive mobilization in patients with sepsis: an observational transversal study. Int J Cardiovasc Imaging 2022, 38, 297-308, doi:10.1007/s10554-021-02402-0.
  4. Rogers, B.; Mourot, L.; Doucende, G.; Gronwald, T. Fractal correlation properties of heart rate variability as a biomarker of endurance exercise fatigue in ultramarathon runners. Physiol Rep 2021, 9, e14956, doi:10.14814/phy2.14956.
  5. Gronwald, T.; Hoos, O.; Hottenrott, K. Effects of a Short-Term Cycling Interval Session and Active Recovery on Non-Linear Dynamics of Cardiac Autonomic Activity in Endurance Trained Cyclists. J Clin Med 2019, 8, doi:10.3390/jcm8020194.

Reviewer 2 Report

The manuscript presents an interesting application of DFA in heart rate variability analysis in anaesthesia. While the topic is important and the paper is well structured, there are some points to be addressed:

-In the title, you mention “biosensor”. A biosensor is a device for the acquisition of biochemical or biological parameters by using chemical reactions or biological interactions. In this context, I would recommend changing “biosensor” to “biomarker”.

-What is the meaning of “ASA I” in line 61?

-Missing “consumption” or “intake” in “Vigorous exercise, alcohol and coffee were prohibited for 48 hours before the scheduled surgery” (lines 68-69).

-Slightly blurred graphs in figures 1-8. I recommend providing sharper versions.

-Not uniform text formatting in tables (compare tables 1 and 2), especially font size. Use one font size for all the tables.

-Figures 5 and 6 are barely legible. If I were you, I would replace the line graphs with box plots.

-Typo in LH/Hf ratio.

Author Response

We thank reviewer for the detailed review and comments, which is responded point by point below:

The manuscript presents an interesting application of DFA in heart rate variability analysis in anaesthesia. While the topic is important and the paper is well structured, there are some points to be addressed:

-In the title, you mention “biosensor”. A biosensor is a device for the acquisition of biochemical or biological parameters by using chemical reactions or biological interactions. In this context, I would recommend changing “biosensor” to “biomarker”.

We follow the reviewer’s recommendation and replace all biosensors with biomarkers

-What is the meaning of “ASA I” in line 61?

The ASA Physical Status Classification System has been in use for over 60 years. The purpose of the system is to assess and communicate a patient’s pre-anesthesia medical co-morbidities and become an international standard for patients’ classification. ASA I is a typo and ASA 1 (healthy patients) is replaced.

-Missing “consumption” or “intake” in “Vigorous exercise, alcohol and coffee were prohibited for 48 hours before the scheduled surgery” (lines 68-69).
All errors are corrected.

-Slightly blurred graphs in figures 1-8. I recommend providing sharper versions.
A sharper version of graphs is provided in the new revision.

-Not uniform text formatting in tables (compare tables 1 and 2), especially font size. Use one font size for all the tables.
All errors are corrected.

-Figures 5 and 6 are barely legible. If I were you, I would replace the line graphs with box plots.
We follow the reviewer’s recommendation and replace the line graphs with box plots for clearer explanation.

-Typo in LH/Hf ratio. 

All errors are corrected.

Round 2

Reviewer 1 Report

None